# GRPO-CARE: Consistency-Aware Reinforcement Learning for Multimodal Reasoning

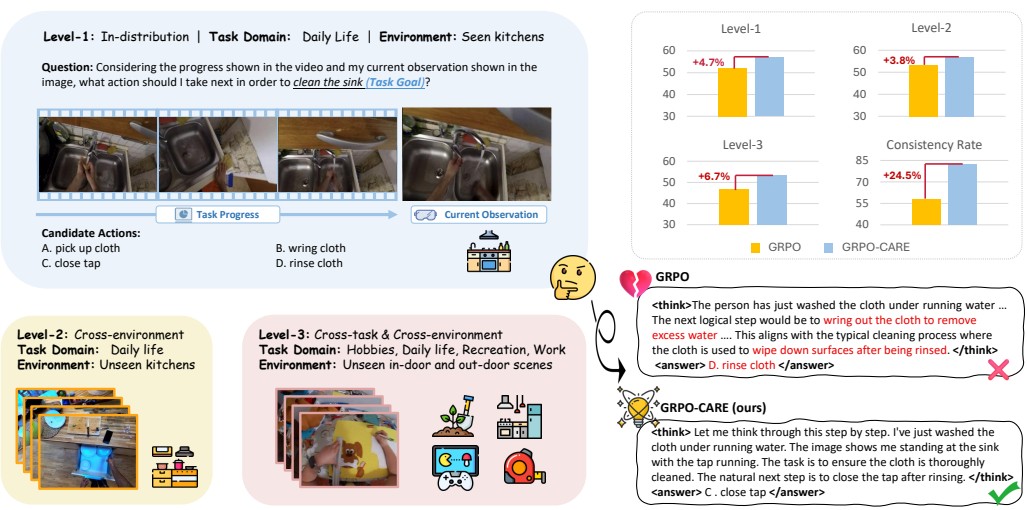

Figure 1: (a) **SEED-Bench-R1** (SB-R1) provides a systematic, three-level evaluation of post-training methods for MLLMs in video understanding, encompassing tasks that require both perception and reasoning to tackle complex real-world scenarios. (b) Our analysis identifies a key limitation of standard outcome-supervised GRPO: while it improves answer accuracy, it often compromises logical consistency between reasoning and answers. By introducing an adaptive, group-relative consistency bonus via reference-likelihood calibration, our **GRPO-CARE** achieves higher answer accuracy across all difficulty levels and improves interpretability, as reflected by increased consistency rates.

## ABSTRACT

Recent reinforcement learning (RL) approaches, such as outcome-supervised GRPO, have advanced Chain-of-Thought reasoning in large language models (LLMs), yet their adaptation to multimodal LLMs (MLLMs) remains underexplored. Progress has been further limited by the lack of evaluation settings that jointly test perception and reasoning under controlled generalization challenges. To enable such analysis, we reorganize prior benchmarks featuring complex real-world videos that demand intricate visual understanding and commonsense planning into **SEED-Bench-R1**, a structured testbed with large-scale training data and hierarchical evaluation across in-distribution, cross-environment, and cross-environment-task scenarios. Using this setting, we conduct a systematic experimental analysis of post-training methods, which reveals a key limitation of outcome-supervised GRPO: while it improves answer accuracy, it often compromises the logical coherence between reasoning and final answers, yielding only a 57.9% consistency rate. This stems from optimizing exclusively for final-answer rewards, which encourages shortcuts, and from rigid KL divergence penalties, which overly constrain adaptive reasoning. To address these issues, we propose **GRPO-CARE**, a novel consistency-aware RL framework that jointly optimizes correctness and coherence without requiring explicit process supervision. GRPO-CARE introduces a two-tiered reward: (1) a base reward for accuracy, and (2) an adaptive consistency bonus derived from a slowly evolving reference model that calibrates reasoning-to-answer likelihoods within peer groups. This mechanism rewards reasoning paths that are both correct and logically consistent, while removing the constraints of KL

penalties. Experiments on SEED-Bench-R1 show that GRPO-CARE consistently outperforms standard GRPO, achieving a 6.7% gain on the hardest evaluation level and a 24.5% increase in reasoning consistency. Moreover, models trained with GRPO-CARE transfer effectively to diverse video understanding and even language-only reasoning benchmarks, highlighting its robustness and generality.

# 1 INTRODUCTION

Recent progress in Large Language Models (LLMs) (Guo et al., 2025; ope, 2024; Team et al., 2025) has been driven by advances in long Chain of Thought (CoT) generation, with reinforcement learning (RL) (Shao et al., 2024; Ouyang et al., 2022; Yeo et al., 2025) emerging as an effective post-training technique that improves complex problem-solving and generalization. Multimodal LLMs (MLLMs) extend these capabilities to process multimodal inputs (Zhang et al., 2025; Liu et al., 2025b; Meng et al., 2025), inheriting strong reasoning abilities while tackling richer, more complex data. However, current evaluations of RL-like post-training for MLLMs are fragmented: some focus narrowly on perception (e.g., detection, grounding) (Liu et al., 2025b), others on logical reasoning (e.g., multimodal math) (Huang et al., 2025), or else rely on broad datasets without structured generalization assessment (Feng et al., 2025).

We argue that *studying post-training for MLLMs requires evaluations that balance perception and logical reasoning, while rigorously testing generalization.* To this end, we reorganize prior benchmarks (Chen et al., 2023; Qiu et al., 2024) featuring complex real-world videos that demand intricate visual understanding and commonsense planning, and construct **SEED-Bench-R1** as a structured testbed. As shown in Fig. 1, SEED-Bench-R1 requires models to comprehend open-form task goals, track long-horizon visual progress, perceive complex environmental cues, and reason about next actions using world knowledge. It features hierarchical evaluation across three levels—(1) in-distribution, (2) OOD cross-environment, and (3) OOD cross-environment-task—with large-scale training data and verifiable ground-truth answers, making it suitable for supporting research in RL-like post-training methods for MLLMs.

Using SEED-Bench-R1, we conduct a comprehensive analysis comparing representative post-training approaches. Our experiments confirm that RL—specifically GRPO with outcome supervision (Shao et al., 2024)—is highly data-efficient and significantly outperforms supervised fine-tuning (SFT) on both in-distribution and OOD questions. However, we identify a key limitation: while outcome-supervised GRPO improves perception and answer accuracy for MLLMs, it often sacrifices logical coherence between reasoning chains and final answers, with a consistency rate of only 57.9%. This restricts interpretability and limits the potential performance ceiling. It originates from the fact that optimizing solely for final answer rewards creates a shortcut, where models prioritize correctness over maintaining logical reasoning, while strict KL penalties hinder adaptive adjustment of causal relations between reasoning and answers.

To overcome this, we propose **GRPO-CARE**, a novel RL framework with **C**onsistency-**A**ware **R**eward **E**nhancement that jointly optimizes answer correctness and logical consistency without relying on explicit process supervision. As illustrated in Fig. 3, in addition to the base reward for answer correctness, we introduce a consistency bonus derived from a slowly-updated reference model through *likelihood calibration*. This bonus incentivizes the model to produce reasoning traces that are not only accurate but also logically coherent with the final answer. Specifically, GRPO-CARE maintains a reference model updated via exponential moving average (EMA) of the online model's parameters, calibrating reasoning-to-answer consistency likelihoods. Samples that achieve both high accuracy and strong consistency are rewarded with an adaptive group-relative bonus, replacing rigid KL penalties and enabling more effective exploration of coherent reasoning paths.

Extensive evaluation shows that GRPO-CARE consistently outperforms standard GRPO across all difficulty levels, especially in challenging OOD scenarios, improving performance by 6.7% on Level-3 and increasing reasoning consistency by 24.5%. Ablation studies confirm that the consistency-aware reward is critical for balancing overall performance and interpretability, while transfer experiments to diverse video understanding benchmarks and even purely language-based reasoning tasks further demonstrate robustness and generality.

In summary, our main contributions are:

- A systematic reorganization of prior benchmarks into **SEED-Bench-R1**, providing a hierarchical and rigorous evaluation setting for studying post-training methods in multimodal reasoning.
- A comprehensive experimental study of post-training methods for MLLMs, revealing the limitations of outcome-supervised RL in maintaining logical coherence.
- A novel RL framework, **GRPO-CARE**, that introduces consistency-aware rewards to significantly improve reasoning interpretability and overall performance without explicit process supervision.

## 2 RELATED WORK

**RL for LLMs/MLLMs.** RL from human feedback (RLHF) aligns LLM outputs with human preferences via reward models trained on human preference data (Ouyang et al., 2022; Schulman et al., 2017). To enhance complex reasoning, generating long CoT is effective (Guo et al., 2025; Team et al., 2025; ope, 2024). RL methods like GRPO (Shao et al., 2024) and its variants DAPO (Yu et al., 2025b) and Dr.GRPO (Liu et al., 2025a) optimize CoT generation using outcome-based rewards. However, outcome-only supervision can yield inconsistent reasoning despite correct answers. Addressing this, some works train additional process supervision reward models with costly step-wise annotations (Lightman et al., 2023; Uesato et al., 2022; Chen et al.; Luo et al., 2024; Wang et al., 2023), incorporate LLM judges (Gao et al., 2024; Xia et al., 2025; Zhang et al., 2024a), or adaptive regularization via EMA-updated reference models (Ramé et al., 2024). In MLLMs, outcome-based RL may cause "Thought Collapse," mitigated by stronger correctors (Wei et al., 2025) or step-wise reward matching (Zhang et al., 2025). Our GRPO-CARE employs a slowly updated reference model to provide bonus feedback on logically consistent and accurate responses, improving reasoning and accuracy without extra annotations or stronger correctors.

**Benchmarks for MLLM Post-training.** Recent RL-based post-training methods for MLLMs have primarily targeted image tasks—from perception (e.g., classification) to reasoning (e.g., visual math) (Huang et al., 2025; Liu et al., 2025b; Zhang et al., 2025; Sun et al., 2024). In contrast, video understanding, a more complex and general scenario, remains underexplored. Early RL-based efforts on video benchmarks (Wang & Peng, 2025; Zhao et al., 2025a) are limited by narrow tasks (e.g., emotion recognition) (Liu et al., 2022; Jiang et al., 2020) or scarce training data (Wu et al., 2024), hindering scalable analysis. Existing benchmarks (Li et al., 2024b; Liu et al., 2024b; Fang et al., 2024) mostly evaluate models post-trained on diverse general-domain data (e.g., Video-R1 (Feng et al., 2025)) but lack rigorous generalization assessment. To date, no comprehensive benchmark provides (1) large-scale training data for robust post-training, (2) structured validation sets across multiple generalization levels, and (3) multimodal questions balancing perception and reasoning in real-world scenarios. To address this, we propose SEED-Bench-R1, a video understanding benchmark with large-scale training data and a validation set partitioned into three generalization tiers, enabling comprehensive evaluation of MLLM post-training methods.

## 3 PILOT STUDY WITH SEED-BENCH-R1

### 3.1 SEED-BENCH-R1

**Benchmark Overview.** As shown in Fig. 1, SEED-Bench-R1 is built to systematically study how post-training methods affect multimodal reasoning in MLLMs. Building on EgoPlan-Bench (Chen et al., 2023) and EgoPlan-Bench2 (Qiu et al., 2024), it features: 1) real-world egocentric visual inputs, 2) diverse questions requiring commonsense reasoning for practical tasks, 3) a hierarchical validation design to assess robustness and generalization, and 4) large-scale, automatically constructed training questions with verifiable ground-truth answers.

**Visual Inputs and Question Design.** SEED-Bench-R1 leverages realistic egocentric videos (Damen et al., 2022; Grauman et al., 2022) of daily activities. Correctly answering its questions requires models to understand open-ended goals, track long-horizon task progress, perceive real-time environment states from an egocentric view, and apply world knowledge to infer the next action. The ground-truth answer comes from the actual next action occurring right after the current observation in the original uncropped video, with the negative options sampled from the same video. This challenging setting of candidate options demands a deep understanding of the environment state from dynamic visual input and world knowledge, such as action dependencies, rather than just semantic matching, to discern the correct action plan. Moreover, the derivation of golden answers is traceable and easy to verify.

Table 1: Statistics of SEED-Bench-R1, including a training set and a hierarchical three-level validation set for in-distribution, cross-environment, and cross-environment-task evaluations.

| Split | # Samples | Domain | Cross-Env. | Cross-Task | Video Source | Benchmark Source |
|-------|-----------|--------|------------|------------|--------------|------------------|
| Train | 50,269 | Daily life | - | - | Epic-Kitchens | EgoPlan-Bench |
| Val-L1 | 2,432 | Daily life | $\times$ | $\times$ | Epic-Kitchens | EgoPlan-Bench |
| Val-L2 | 923 | Daily life | $\sqrt{}$ | $\times$ | Ego4D | EgoPlan-Bench |
| Val-L3 | 1,321 | Hobbies, Daily life, Recreation, Work | $\sqrt{}$ | $\sqrt{}$ | Ego4D | EgoPlan-Bench2 |

**Dataset Composition and Validation Levels.** As shown in Tab. 1, SEED-Bench-R1 includes both training and validation sets. The training set is automatically generated from Epic-Kitchens (Damen et al., 2022), covering daily household tasks. The validation set is human-verified and divided into three levels: **L1**: In-distribution evaluation with the same source and domain as training. **L2**: Cross-environment evaluation, using unseen kitchen environments from Ego4D (Grauman et al., 2022). **L3**: Cross-environment-task evaluation with the full Ego4D set, spanning hobbies, recreation, work, and daily life in diverse indoor and outdoor contexts, thus testing broader generalization.

### 3.2 EXPERIMENT SETUP

We use Qwen2.5-VL-Instruct-7B (Bai et al., 2025) as the backbone to study post-training on SEED-Bench-R1. Additional evaluation on different base models can be found in Appendix B.2. We adopt outcome-supervised GRPO (Shao et al., 2024) as a representative RL method and compare it with supervised finetuning (SFT). Each video is down-sampled to 16 frames at resolution $128 \times 28 \times 28$, plus one frame indicating the current observation. For SFT, training data is augmented with CoT reasoning distilled from Qwen2.5-VL-Instruct-72B and 7B via rejection sampling. GRPO instead relies on rule-based rewards without explicit CoT annotations. Following Guo et al. (2025), reasoning and answers are formatted within `<think>` `</think>` and `<answer>` `</answer>` tags.

Given a multimodal question $x \sim \mathcal{D}$, GRPO samples $G$ responses $\{o_g = (\tau_g, a_g)\}_{g=1}^{G}$ from the policy $\pi_{\theta_{\text{old}}}$, where $\tau_g$ and $a_g$ denote reasoning and answer. Unlike SFT, GRPO does not rely on predefined responses. The policy is optimized by maximizing:

$$\mathcal{J}_{\text{GRPO}}(\theta) = \mathbb{E}_{x, \{o_g\}} \frac{1}{G} \sum_{g=1}^{G} \frac{1}{|o_g|} \sum_{i=1}^{|o_g|} \min \left[ \frac{\pi_\theta(o_{g,i}|x, o_{g,<i})}{\pi_{\theta_{\text{old}}}(o_{g,i}|x, o_{g,<i})} \hat{A}_{g,i}, \right.$$

$$\left. \text{clip}\left( \frac{\pi_\theta(o_{g,i}|x, o_{g,<i})}{\pi_{\theta_{\text{old}}}(o_{g,i}|x, o_{g,<i})}, 1-\varepsilon, 1+\varepsilon \right) \hat{A}_{g,i} \right] - \beta \mathbb{D}_{KL}[\pi_\theta||\pi_{\text{ref}}]$$

Here, $\varepsilon$ and $\beta$ are hyperparameters, and $\mathbb{D}_{KL}$ is the KL divergence between policy $\pi_\theta$ and reference $\pi_{\text{ref}}$. The per-token advantage is set to the normalized reward $\hat{A}_{g,i} = \widetilde{r}_g = \frac{r_g - \text{mean}(\{r_1, ..., r_G\})}{\text{std}(\{r_1, ..., r_G\})}$, with $r_g$ computed by rules (e.g., $r_g = 1$ if the extracted answer matches ground truth, else 0).

### 3.3 RESULT ANALYSIS

Tab. 2 summarizes the performance of MLLMs trained with various methods on SEED-Bench-R1. Notably, compared to SFT, reinforcement learning with GRPO significantly improves data efficiency and boosts MLLM performance on both in-distribution (L1) and OOD (L2, L3) questions, despite relying only on a simple outcome-based reward without specialized CoT annotations.

Our analysis shows that GRPO mainly enhances perceptual abilities rather than reasoning. As shown in Fig. 2, the SFT-trained model is more prone to perceptual hallucinations, such as describing "a ball being hit from a tee" when this event does not occur. Attention map analysis reveals that GRPO-trained models generate CoT tokens that act as dynamic queries, attending to visual content more thoroughly—especially in OOD scenarios. For example, the GRPO model better highlights key visual observations and allocates more attention to critical objects (e.g., the ball on the tee), even if these are not explicitly referenced in the reasoning. We hypothesize that RL methods like GRPO encourage broader visual exploration via CoT, while SFT tends to produce superficial, pattern-memorized CoT with limited visual grounding. This likely underpins GRPO's superior generalization.

However, outcome-supervised GRPO training for MLLMs has key limitations: unlike LLMs, MLLM reasoning does not improve proportionally during RL, often resulting in logical inconsistencies.

Table 2: Performance comparison on SEED-Bench-R1's hierarchical validation set.

| Models | L1 (In-Distribution) | L2 (Cross-Env) | L3 (Cross-Task, Cross-Env) | | | | |
|---|---|---|---|---|---|---|---|
| | Daily Life | Daily Life | Daily Life | Hobbies | Recreation | Work | Overall |
| Qwen2.5-VL-7B | 38.4 | 40.1 | 35.8 | 31.2 | 26.8 | 28.5 | 31.3 |
| SFT | 46.2 | 46.3 | 46.7 | 41.7 | 44.3 | 38.4 | 42.7 |
| GRPO | 52.3 | 53.2 | 51.9 | 43.7 | 55.2 | 39.4 | 46.7 |
| GRPO-CARE (ours) | **57.0** | **57.0** | **57.6** | **51.2** | **57.4** | **48.5** | **53.4** |

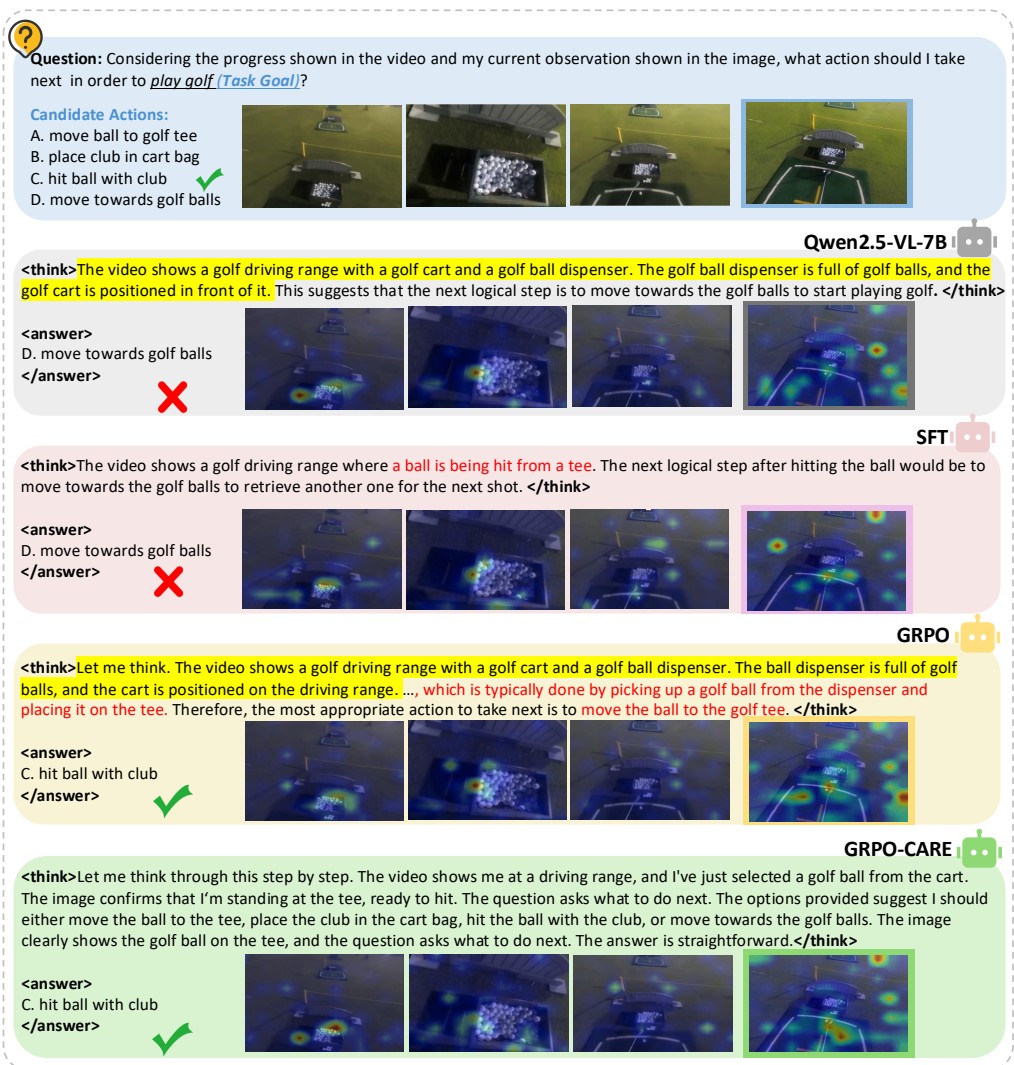

Figure 2: Case study of an L3 question from SEED-Bench-R1, showing a video of task progress, a final observation image, and attention maps (output-to-visual tokens). The SFT model tends to memorize reasoning patterns and exhibits perceptual hallucinations. The GRPO model attends more comprehensively to the highlighted key visual observation while lacking logical consistency in the generated content. The GRPO-CARE model further balances visual perception and logical reasoning.

While the GRPO-trained model frequently reaches correct answers, its CoT reasoning often lacks coherence. For instance, as shown in Fig. 2, initial reasoning steps mirror those of the base model (Qwen2.5-VL-7B), but later steps diverge and may contradict each other—e.g., suggesting "move the ball to the golf tee" but ultimately answering "hit ball with club." Such inconsistencies, though sometimes yielding correct answers, undermine transparency.

Limited reasoning also constrains overall performance, as effective reasoning is crucial for integrating world knowledge with perception. For example, in Fig. 1, the GRPO model correctly identifies

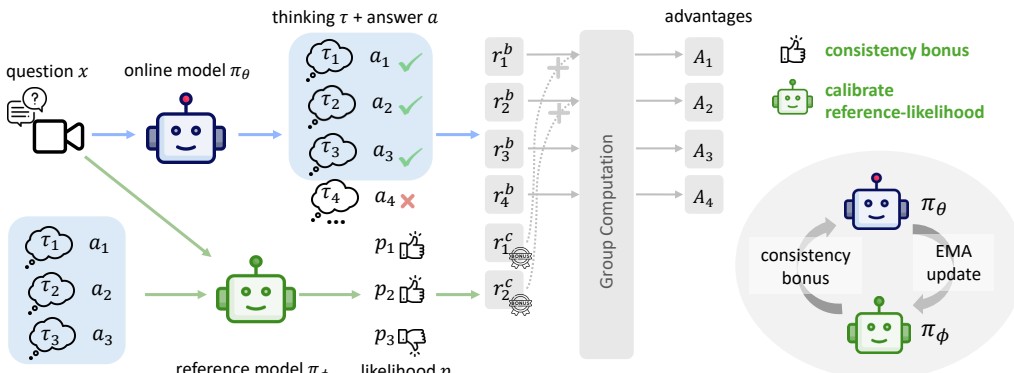

Figure 3: GRPO-CARE uses a two-tier reward system: a base reward for answer correctness ($r_*^b$) and an adaptive consistency bonus ($r_*^c$). The consistency bonus is given to high-accuracy samples whose reasoning-to-answer likelihood—estimated by a slowly updated (EMA) reference model—is higher than that of their group peers, conditioned on the multimodal question. The total reward, the sum of base and consistency rewards, is then used to compute advantages for updating the online model.

"running water" but fails to infer that the next logical step after cleaning is "turning off the faucet." These reasoning-answer mismatches further complicate interpretability.

## 4 CONSISTENCY-AWARE REWARD-ENHANCED GRPO (GRPO-CARE)

While outcome-supervised GRPO enhances visual perception in MLLMs, our analysis on SEED-Bench-R1 uncovers a critical trade-off: it often produces less logically coherent reasoning chains, thereby limiting interpretability and performance. This issue arises from two main limitations. First, the standard reward focuses exclusively on final-answer accuracy, overlooking the quality of intermediate reasoning steps. This can incentivize shortcut solutions—correct answers reached via inconsistent reasoning. Second, the KL penalty disproportionately constrains reasoning traces, typically longer than answers, thereby stifling exploration of diverse and coherent reasoning paths.

To address these challenges, we propose **GRPO-CARE** (**C**onsistency-**A**ware **R**eward **E**nhancement), a method that jointly optimizes for both answer correctness and logical consistency, without requiring explicit supervision of the reasoning process. As shown in Fig. 3, GRPO-CARE introduces a two-tiered reward system: a base reward for answer correctness, and an adaptive consistency bonus. The consistency bonus is calculated by comparing the likelihood that a reasoning trace leads to the correct answer, as estimated by a slowly evolving reference model. For each high-accuracy sample generated by the online model, this likelihood is compared with those of its peers within the same group, encouraging the exploration of reasoning traces that are logically consistent with correct answers.

The training process, detailed in Algorithm 1, involves **two-stage filtering**. (1) First, we generate multiple reasoning traces per input and retain only those that exceed an accuracy baseline. (2) For these high-accuracy candidates, we assess how well each reasoning trace supports the final answer by calibrating its likelihood using a slowly evolving reference model.

**Reference Model and Likelihood Calibration.** The key insight is that *a stable reference model—when conditioned on the online model's reasoning trace—should assign a higher likelihood to the correct answer if the reasoning is logically grounded in the multimodal input*. Specifically, the reference model is initialized from the same pretrained weights as the online model and updated smoothly via exponential moving average (EMA) to ensure stable likelihood estimation and self-adaptation. This setup encourages gradual self-evolution from a weaker to a stronger model without relying on a stronger external teacher, unlike methods that depend on fixed, stronger LLM judges. To avoid reinforcing "consistent-but-wrong" reasoning, we compute this likelihood only for trajectories with correct answers. Additionally, we cap the likelihood at a maximum threshold to prevent over-optimization toward artificially high values.

**Consistency Bonus Calculation.** Based on the clipped reference likelihoods, we compute a *group-relative consistency baseline* as the mean clipped likelihood (minus a small margin to avoid penalizing

---

**Algorithm 1** Consistency-Aware Reward Enhanced GRPO

---
**Require:**
    $\pi_\theta$: Online policy model (initialized from pretrained weights)
    $\pi_\phi$: Reference model with EMA updates ($\phi \leftarrow \theta$ initially)
    $\mathcal{D}$: Multimodal training dataset $\{(x, y^*)\}$
    $\lambda_{\text{cons}}$: Consistency reward coefficient (e.g., 0.5)
    $\gamma_{\text{acc}}$: Minimum accuracy threshold (e.g. 0.1)
    $\gamma_p$: Maximum likelihood threshold (e.g. 0.95)
    $\epsilon_p$: Consistency margin (e.g. 0.01)
1: **procedure** TRAINING($\pi_\theta, \mathcal{D}, T$)
2:    **for** $t \leftarrow 1$ to $T$ **do**
3:        **for** each multimodal input $x$ in batch $\mathcal{D}$ **do**
4:            **Phase 1: Trajectory Generation & Reward Computation**
5:            Generate $G$ reasoning traces + answers: $\{\tau_g, a_g\}_{g=1}^G \sim \pi_\theta(\cdot|x)$
6:            Compute accuracy rewards: $r_{\text{acc},g} = \text{accuracy\_score}(a_g, y^*)$
7:            Compute format rewards: $r_{\text{fmt},g} = \text{format\_score}(\tau_g, a_g)$
8:            **Phase 2: Relative High-Accuracy Trajectory Selection**
9:            Calculate relative accuracy baseline: $\hat{r}_{\text{acc}} = \max(\mathbb{E}_g[r_{\text{acc},g}], \gamma_{\text{acc}})$
10:           Select trajectories where $r_{\text{acc},g} \geq \hat{r}_{\text{acc}}$
11:           **Phase 3: Relative Consistency Evaluation**
12:           **for** selected trajectories $(\tau_g, a_g)$ **do**
13:               Compute reference likelihood: $p_g = \frac{1}{|a_g|} \sum_{i=1}^{|a_g|} \pi_\phi(a_{g,i} \mid x, \tau_g, a_{g,<i})$
14:               Clip likelihood: $\tilde{p}_g = \min(p_g, \gamma_p)$
15:           **end for**
16:           Calculate relative consistency baseline: $\hat{\mu}_p = \mathbb{E}_g[\tilde{p}_g] - \epsilon_p$
17:           Select consistent trajectories where $\tilde{p}_g \geq \hat{\mu}_p$
18:           **Phase 4: Enhanced Reward Calculation**
19:           **for** each trajectory $g$ **do**
20:               $R_g = \underbrace{r_{\text{acc},g} + r_{\text{fmt},g}}_{\text{base reward}} + \underbrace{\lambda_{\text{cons}} \cdot r_{\text{acc},g} \cdot \mathbb{I}[\text{consistent}]}_{\text{consistency bonus}}$
21:               Normalize advantage: $\hat{A}_g = (R_g - \mu_R)/\sigma_R$
22:           **end for**
23:        **end for**
24:        **Phase 5: Model Update**
25:        Update $\pi_\theta$ via GRPO policy gradient (without KL penalty)
26:        **if** $t \bmod k = 0$ **then**                       ▷ EMA update every $k = 10$ steps
27:           $\phi \leftarrow \alpha\phi + (1-\alpha)\theta$                     ▷ $\alpha$=0.995 typical
28:        **end if**
29:    **end for**
30:    **return** optimized policy $\pi_\theta$
31: **end procedure**

---

near-average samples). Trajectories that exceed this baseline receive a sparse *consistency bonus*, weighted by their accuracy, ensuring that rewards prioritize both correctness and logical coherence.

**Model Update.** To promote exploration of diverse reasoning paths, we remove the KL penalty from the GRPO's training objective. Instead, we rely on the consistency bonus—added to the base reward to form the total reward—to guide online model updates toward higher-quality outputs. The reference model is updated via EMA every few steps, allowing it to gradually inherit improvements from the online model (e.g., better visual grounding or more complex reasoning) while maintaining stability against sampling noise. This balanced optimization process enhances multimodal understanding without sacrificing logical consistency, ultimately improving both performance and interpretability.

## 4.1 EVALUATION ON SEED-BENCH-R1

We first evaluate our method on SEED-Bench-R1. As shown in Tab. 2, GRPO-CARE significantly outperforms GRPO across all three difficulty levels, with a particularly notable improvement of nearly 10% on the most challenging L3 evaluation in domains such as *Hobbies* and *Work*, which exhibit substantial distributional divergence from the training data.

To thoroughly assess the effectiveness of GRPO-CARE, we compare it against two families of baseline methods: **KL-oriented baselines**, which modify the application of divergence constraints, and **reward-based alternatives**, which replace KL penalties with consistency-aware rewards.

- **KL-Oriented Baselines. 1) KL-EMA** introduces an EMA-updated reference model for adaptive constraints, following the implementation of WARP (Ramé et al., 2024). **2) KL-EMA-HA** selectively applies KL penalty only to high-accuracy samples, applying regularization on where alignment matters most. **3) SepKL-EMA-HA** further decomposes KL into separate terms for reasoning and answer tokens to alleviate disproportionately penalizing lengthy reasoning tokens while potentially overlooking answer-reasoning inconsistencies. **4) NoKL** removes the KL penalty, demonstrating the raw optimization potential absent any regularization.
- **Reward-Based Alternatives. 5) DenseCons** applies continuous likelihood weighting to derive dense consistency rewards: $r_{\text{cons}} = \lambda_{\text{cons}} \cdot r_{\text{acc}} \cdot p_{\phi}(a|x, \tau)$. **6) RefGen** takes a more explicit approach by having the reference model regenerate answers from sampled reasoning paths, using the regenerated answer's accuracy as the consistency signal: $r_{\text{cons}} = \lambda_{\text{cons}} \cdot \text{accuracy}(a' \sim \pi_{\phi}(\cdot|x, \tau), y^*)$.

As shown in Tab. 3, we report both benchmark performance and the consistency rate between generated reasoning and final answers, where consistency is evaluated by GPT-4.1 to assess whether the reasoning sufficiently supports the answer. The details for consistency evaluation is described in Appendix B.3. Our analysis shows that while the EMA-updated reference model improves both accuracy and consistency, restricting KL penalties to high-accuracy samples (*KL-EMA-HA*) boosts in-domain (L1) results but slightly reduces OOD (L2/L3) general-

Table 3: Ablation results of KL-oriented and reward-based baselines on SEED-Bench-R1.

| Models | L1 | L2 | L3 | Consistency |
|---|---|---|---|---|
| GRPO | 52.3 | 53.2 | 46.7 | 57.9 |
| KL-EMA | 54.7 | 54.1 | 49.4 | 60.0 |
| KL-EMA-HA | 55.1 | 53.8 | 49.2 | 61.7 |
| SepKL-EMA-HA | 54.8 | 54.9 | 47.5 | 76.8 |
| NoKL | 55.4 | 54.4 | 51.3 | 70.0 |
| DenseCons | 56.6 | 55.5 | 50.6 | 80.3 |
| RefGen | 55.2 | 54.2 | 49.4 | **86.4** |
| CARE (ours) | **57.0** | **57.0** | **53.4** | 82.4 |

ization. Decomposing KL penalties (*SepKL-EMA-HA*) mitigates reasoning-answer inconsistency, yielding minor gains on L2 but limited impact on L3. Notably, none of the KL-based variants outperform *NoKL*, indicating that standard KL regularization may hinder the performance ceiling in this context.

Among reward-based methods, *DenseCons* surpasses *NoKL* on L1 and L2 with improved consistency, but slightly underperforms on L3, likely due to over-reliance on reference model calibration. *RefGen* greatly increases consistency but introduces instability from sampling-based answer regeneration, ultimately reducing overall performance. Our proposed **GRPO-CARE** uses sparse consistency rewards to achieve robust improvements across all levels. Its two-stage filtering—leveraging adaptive EMA-updated reference likelihoods to provide relative, sparse feedback for high-accuracy samples—effectively enhances logical consistency and answer accuracy. This demonstrates that group-relative sparse rewards deliver more reliable learning signals, avoiding overfitting to imperfect likelihoods (as in DenseCons) or sampling noise (as in RefGen).

**Effect of the EMA Reference Model.** To investigate the role of the EMA reference model in consistency assessment for reward shaping, we conduct additional analysis in Tab. 4. Using an EMA-updated reference yields significantly better performance than a frozen reference, suggesting that EMA effectively adapts the reference model to the evolving reasoning process of the online policy. Importantly, the EMA-updated reference does not simply converge to the online policy: per-

Table 4: Impact of different reference models on GRPO-CARE.

| Reference | L1 | L2 | L3 |
|---|---|---|---|
| Frozen | 51.6 | 50.4 | 51.1 |
| EMA updated | **57.0** | **57.0** | **53.4** |
| Online policy | 52.7 | 53.5 | 51.2 |

formance with the EMA reference surpasses that of using the online policy itself, confirming that it remains a distinct and stable reference.

**Hyperparameter Sensitivity.** We use the same hyperparameters across different benchmarks: EMA update every 10 steps, EMA decay 0.995, likelihood cap 0.95, and consistency margin 0.01. We further validate the robustness of GRPO-CARE by ab-

Table 5: Ablation results on key hyperparameters.

| Hyperparameter | Values | Performance |
|---|---|---|
| Consistency margin | 0.0 / 0.01 / 0.05 / 0.1 | 56.2 / **57.0** / 56.6 / 50.5 |
| Likelihood cap | 1.0 / 0.99 / 0.95 / 0.90 | 54.0 / 56.1 / **57.0** / 56.0 |
| EMA frequency | 1 / 10 / 50 / $\infty$ | 54.1 / **57.0** / 56.8 / 51.6 |
| EMA decay | 0 / 0.99 / 0.995 / 0.999 | 52.9 / 56.1 / **57.0** / 52.5 |

Table 6: Performance of different models on general video understanding benchmarks

| Models | VSI-Bench | VideoMMMU | MMVU | MVBench | TempCompass | VideoMME |
|---|---|---|---|---|---|---|
| GPT-4o (Hurst et al., 2024) | 34.0 | 61.2 | 75.4 | - | - | 71.9 |
| LLaMA-VID (Li et al., 2024c) | - | - | - | 41.9 | 45.6 | - |
| VideoLLaMA2 (Cheng et al., 2024) | - | - | 44.8 | 54.6 | - | 47.9 |
| LongVA-7B (Zhang et al., 2024b) | 29.2 | 23.9 | - | - | 56.9 | 52.6 |
| VILA-1.5-8B (Lin et al., 2024) | 28.9 | 20.8 | - | - | 58.8 | - |
| Video-UTR-7B (Yu et al., 2025a) | - | - | - | 61.1 | 62.5 | 56.0 |
| LLaVA-OneVision-7B (Li et al., 2024a) | 32.4 | 33.8 | 49.2 | 56.7 | - | 58.2 |
| Kangeroo-8B (Liu et al., 2024a) | - | - | 61.1 | 62.5 | 69.9 | 55.4 |
| Video-R1-7B (Feng et al., 2025) | **35.8** | **52.3** | 63.8 | 63.9 | 73.2 | 59.3 |
| Qwen2.5-VL-7B | 30.1 | 48.1 | 60.0 | 59.0 | 72.6 | 56.6 |
| CARE-7B (SB-R1) | 34.3 | 51.6 | **66.2** | 63.2 | **74.3** | 58.1 |
| CARE-7B (Video-R1) | **35.8** | 50.4 | 65.8 | **65.1** | 73.5 | **59.6** |

lating key hyperparameters. Table 5 reports L1 performance on SEED-Bench-R1 under different settings, confirming that our default choices are stable and effective.

## 4.2 GENERALIZATION AND TRANFERABILITY BEYOND SEED-BENCH-R1

To comprehensively evaluate our model's capabilities, we conduct extensive experiments on both general video understanding and language-only reasoning benchmarks beyond SEED-Bench-R1.

For video understanding, we benchmark our model on six challenging datasets spanning diverse aspects: spatial reasoning (VSI-Bench (Yang et al., 2024)), knowledge-intensive QA (VideoMMMU (Hu et al., 2025) and MMVU (Zhao et al., 2025b)), and general video understanding (MVBench (Li et al., 2024b), TempCompass (Liu et al., 2024b), and VideoMME (Fu et al., 2024)). As shown in Tab. 6, our CARE-7B (SB-R1) achieves significant performance improvements over the base model across all benchmarks after training on SEED-Bench-R1. These consistent gains validate the quality of our benchmark's training data, the robustness of our methodology, and the comprehensiveness of our evaluation protocol. Furthermore, we conduct additional experiments following Video-R1 (Feng et al., 2025), training our model using GRPO-CARE with 16-frame video inputs on general-domain data (Video-R1-260k) for 1k RL steps and testing with 32-frame inputs. The comparative results from other baselines are taken from the Video-R1 paper. Notably, even when trained solely with RL, our model achieves competitive or superior performance compared to Video-R1-7B on most benchmarks. This is particularly remarkable given that Video-R1-7B benefits from explicit temporal order grounding constraints via GRPO rewards and supplementary supervised fine-tuning with additional data. Our model's ability to match or outperform this strong baseline with a more streamlined training pipeline underscores the efficiency of our method.

Beyond video understanding, we further evaluate whether our approach generalizes to purely language-based reasoning tasks. Specifically, we test Qwen2.5-VL-7B on GSM8k (Cobbe et al., 2021) and GPQA (Rein et al., 2024) benchmarks, comparing the base model with variants trained using GRPO and GRPO-CARE on SEED-Bench-R1. As reported in Tab. 7, GRPO-CARE yields consistent improvements over both the

Table 7: Performance on language-only reasoning benchmarks.

| Method | GSM8k | GPQA |
|---|---|---|
| Qwen2.5-VL-7B | 72.10 | 28.79 |
| GRPO | 74.07 | 24.24 |
| GRPO-CARE | **81.58** | **30.81** |

base model and vanilla GRPO. Interestingly, while GRPO alone slightly degrades performance on GPQA, incorporating CARE effectively reverses this trend and produces consistent gains. These results suggest that our method not only strengthens multimodal reasoning but also enhances general reasoning capabilities in purely textual domains.

## 5 CONCLUSION

In this paper, we introduced SEED-Bench-R1, a structured benchmark for evaluating post-training methods for MLLM, and proposed GRPO-CARE, a novel consistency-aware RL framework. Our analysis shows that while outcome-supervised GRPO improves accuracy, it often sacrifices reasoning coherence. GRPO-CARE addresses this by rewarding both correctness and consistency using likelihood calibration, leading to stronger generalization, higher interpretability, and effective transfer across tasks. We envision SEED-Bench-R1 and GRPO-CARE as useful tools for advancing robust post-training methods, driving the development of more powerful MLLMs.

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

## A    DETAILS OF SEED-BENCH-R1

The questions from SEED-Bench-R1 are presented as multiple-choice problems and organized into a three-level hierarchy: Level 1 (in-distribution), Level 2 (cross-environment), and Level 3 (cross-task-environment). Figure 4 shows example questions from each level, where the model is required to reason about the next appropriate action using world knowledge, based on the specified task goal and the visual inputs showing task progress and current observation. Specifically, Levels 1 and 2 focus on daily-life household tasks similar to those in the training data. Level 1 questions are set in environments seen during training, while Level 2 questions are set in new, previously unseen environments. Level 3 is the most challenging, covering a broader range of task domains—including Work, Hobbies, and Recreation, as well as Daily Life—and takes place in a wider variety of unseen indoor and outdoor environments. The complete annotation files are included in the attachment, and we will make the corresponding videos publicly available after the release of this paper.

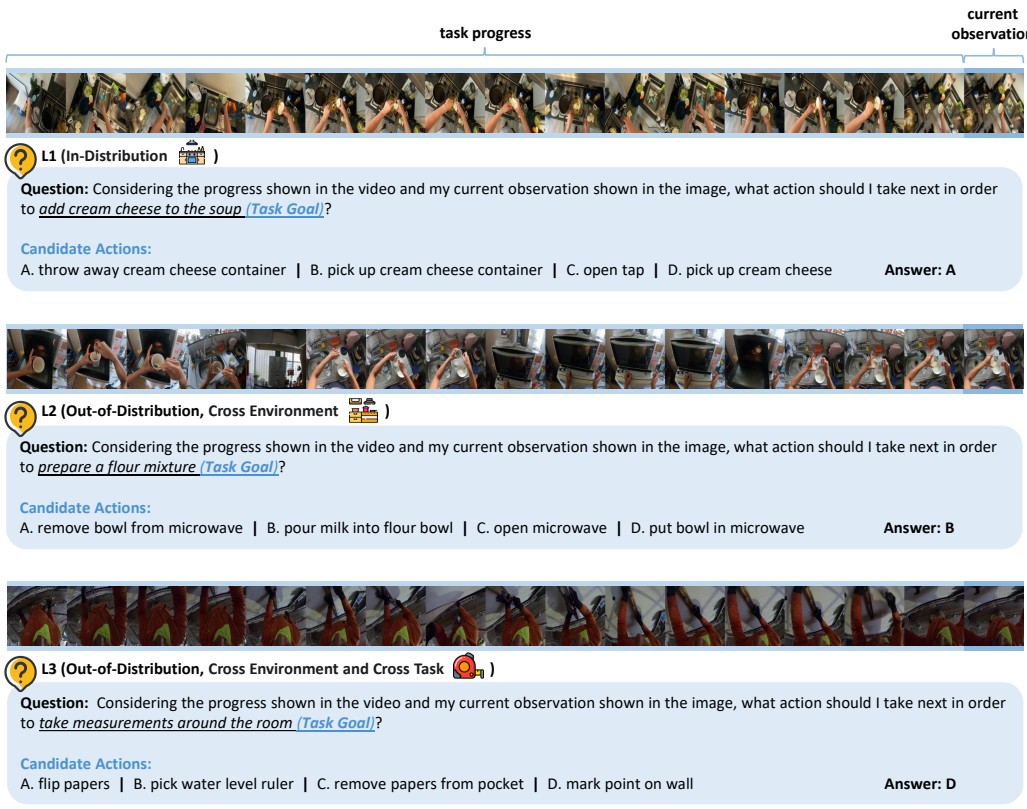

Figure 4: Example questions from the three-level evaluation hierarchy in SEED-Bench-R1's validation set, including in-distribution, cross-environment, and cross-environment-task scenarios.

## B    EXPERIMENTAL DETAILS

### B.1    COMPUTE RESOURCES

For both SFT and GRPO, we utilize four 80GB GPUs with a batch size of 4. The generation group size for GRPO is set to 8 per sample. To improve training efficiency, the number of video frames is limited to 16, with each frame resized to a resolution of $128 \times 28 \times 28$. In the experiments on SEED-Bench-R1, we train the model using 6k out of 50k samples from SEED-Bench-R1's training data for the pilot study. For experiments involving Video-R1-260k, we follow the protocol in Video-R1 (Feng et al., 2025) to train the model for 1,000 steps using a subset of the data. The training time is about 8 hours for GRPO and 50 minutes for SFT.

## B.2 ADDITIONAL EVALUATION ON DIFFERENT BASE MODELS

To further validate the generality of our method, we conduct experiments with GRPO-CARE on multiple Qwen variants of different scales. Recent RL-based post-training studies for MLLMs (Liu et al., 2025b; Feng et al., 2025; Deng et al., 2025) commonly adopt Qwen series models due to their strong open-source ecosystem and competitive performance on multimodal benchmarks. Following this trend, we select Qwen2.5-VL-3B and Qwen2-VL-7B as alternative base models in our evaluation.

As shown in Tab. 8, GRPO-CARE consistently surpasses both SFT and vanilla GRPO across all tested variants. The performance improvements are not only observed on larger models like Qwen2-VL-7B, but also on smaller-scale ones such as Qwen2.5-VL-3B, indicating that our approach is both robust and scalable.

Table 8: Performance of various MLLMs on SEED-Bench-R1. GRPO-CARE consistently improves performance across different base models and model sizes.

| Model | L1 | L2 | L3 |
|---|---|---|---|
| BLIP-2 (Li et al., 2023) | 26.4 | 27.3 | 26.2 |
| InstructBLIP (Dai et al., 2023) | 27.0 | 25.9 | 26.3 |
| Valley (Luo et al., 2023) | 26.8 | 26.8 | 27.0 |
| Yi-VL (Young et al., 2024) | 30.0 | 29.9 | 23.5 |
| LLaVA1.5 (Liu et al., 2023) | 30.7 | 31.0 | 25.4 |
| DeepSeek-VL (Lu et al., 2024) | 31.5 | 33.2 | 28.5 |
| VideoLLaMA3-7B (Zhang et al., 2023) | 33.3 | 33.2 | 27.7 |
| InternVL3-8B (Zhu et al., 2025) | 38.0 | 37.8 | 31.8 |
| Qwen2-VL-7B (Wang et al., 2024) | 34.7 | 34.0 | 31.6 |
| Qwen2-VL-7B + SFT | 43.8 | 44.1 | 38.2 |
| Qwen2-VL-7B + GRPO | 46.0 | 50.2 | 44.9 |
| Qwen2-VL-7B + GRPO-CARE | 57.2 | 56.2 | 53.8 |
| Qwen2.5-VL-3B (Bai et al., 2025) | 31.3 | 32.7 | 28.2 |
| Qwen2.5-VL-3B + SFT | 35.9 | 39.1 | 33.7 |
| Qwen2.5-VL-3B + GRPO | 39.6 | 41.0 | 35.4 |
| Qwen2.5-VL-3B + GRPO-CARE | 47.1 | 48.8 | 43.5 |

For context, we also report the performance of several representative MLLMs on SEED-Bench-R1 in Tab. 8. These results serve as reference baselines to illustrate the overall performance range of current models. Notably, GRPO-CARE consistently improves Qwen variants beyond their supervised and vanilla GRPO counterparts, demonstrating potential for extension to broader multimodal architectures.

## B.3 CONSISTENCY EVALUATION

We use GPT-4.1 to assess the consistency between the model's reasoning process and its final answer. The detailed prompt is shown in Figure 5, where GPT-4.1 is instructed to analyze the logical flow of the model's response and assign a score reflecting whether the reasoning supports the final answer.

```
Question:{}
Procedure:{}
Answer:{}

The above is a **Question**, along with a model's Solution **Procedure** and the **Answer**.

Please fulfill the following requirements:
1. **Consistency Analysis**. Check if the solution process supports the answer and if the answer can be logically derived
from the process. If consistent, the score is 1, with reasons provided.
2. **Logic Flow Analysis**. For example, derive A to get answer X; refine step A to verify the correctness of answer X.
3. **Other Analysis**. Identify any other issues in the solution steps and name them to alert the solver.

Reply in JSON format:
```json
{{"consistency_analysis":{{"score": 0 or 1, "reasons":""}}, "logic_flow":"analysis of procedure", "others":""}}
```

There is no need to consider whether the answer is correct; just analyze the above indicators.
```

Figure 5: The detailed prompt for evaluating consistency between the reasoning process and the final answer using GPT-4.1.

To validate GPT-4.1's judgments, we randomly sampled 50 consistent and 50 inconsistent cases flagged by GPT-4.1 and compared them with human annotations, achieving a Pearson correlation of 0.82. This demonstrates that GPT-4.1 provides reliable and nuanced evaluations of consistency. Below we present example GPT-4.1 evaluations to illustrate the evaluation process.

**Example 1.**

```
{
  "task_goal": "prepare the teapot with tea leaves",
  "options": [
    "A. put down teabag holder",
    "B. scoop tea leaves",
    "C. take bag of tea",
    "D. open bag of tea"
  ],
  "model_response": "<think>Let me think. The person has already opened
      the bag of tea and is now reaching for the teapot. This suggests
      that they are in the process of preparing the teapot with tea
      leaves. The next logical step would be to scoop the tea leaves into
       the teapot. Therefore, the most appropriate action would be to
      scoop the tea leaves.</think><answer>B</answer>",
  "gpt_evaluation": {
    "consistency_analysis": {
      "score": 1,
      "reasons": "The procedure describes reaching for the teapot after
          opening the bag of tea, and logically concludes that the next
          action should be to scoop tea leaves. This directly supports
          the chosen answer (B: scoop tea leaves)."
    },
    "logic_flow": "The procedure starts from the current observation (bag
        of tea opened, reaching for teapot), and infers that scooping
        tea leaves is the next step in preparing the teapot, which
        matches the answer given. Each reasoning step directly leads to
        the selected answer.",
    "others": "No other major issues identified. The reasoning is clear
        and follows from the initial observations and procedural logic."
  }
}
```

**Example 2.**

```
{
  "task_goal": "wash knife, cutting board, and peeler",
  "options": [
    "A. put cutting board onto drying rack",
    "B. put knife onto drying rack",
    "C. pick up cutting board",
    "D. adjust tap"
  ],
  "model_response": "<think>The person has just adjusted the tap, which
      suggests they may have finished washing the cutting board. The
      cutting board is currently resting on the drying rack, which
      implies it may be ready to be washed.</think><answer>A</answer>",
  "gpt_evaluation": {
    "consistency_analysis": {
      "score": 0,
      "reasons": "The procedure states that the cutting board is already
          resting on the drying rack, which implies that action A (put
          cutting board onto drying rack) is redundant or already
          completed. Therefore, the process does not support the answer,
          and it cannot be logically derived from the provided reasoning
          ."
    },
    "logic_flow": "The reasoning chain concludes that the cutting board
        is already on the drying rack, so logically the next step should
        not be to put it there again. Instead, attention should shift to
        another item (knife or peeler) or another action. This breaks the
         logic flow from observation to answer.",
    "others": "There is confusion in the think step: it simultaneously
        claims that the cutting board is on the drying rack and that it
```

```
        may be ready to be washed, which are contradictory states.
        Clearer distinction between current and next state is needed."
  }
}
```

## C  LIMITATIONS

While our work introduces SEED-Bench-R1 and the GRPO-CARE framework to advance post-training for MLLMs, several limitations remain. (1) Although SEED-Bench-R1 provides a rigorous benchmark for video understanding with hierarchical evaluation, it does not yet encompass all possible multimodal domains or modalities. Expanding its coverage in future iterations will be important for further validating the generality of models. (2) The consistency-aware reward in GRPO-CARE relies on model-internal likelihoods and group calibration, which, despite their effectiveness, may not fully capture subtle reasoning errors or always align with human judgment. (3) Similar to other RL-based frameworks, our method incurs additional computational costs due to the need for maintaining reference models and performing group-based calibration. While this overhead was manageable in our experiments, scaling to larger models or more complex tasks may necessitate further optimization. We believe these aspects open valuable opportunities for further research built upon our contributions.

## D  BROADER IMPACTS

On the positive side, the proposed consistency-aware reinforcement learning framework, GRPO-CARE, enhances transparency in AI systems by encouraging logically coherent reasoning chains. This is particularly important in domains such as education, healthcare, and assistive robotics, where understanding the reasoning process is essential. Meanwhile, by mitigating shortcut learning, our approach improves generalization to real-world scenarios, paving the way for safer and more reliable AI applications. Moreover, the release of SEED-Bench-R1 alongside our training framework provides valuable tools to support ongoing research in multimodal understanding.

Nonetheless, potential negative societal impacts warrant careful consideration. Although our method enhances reasoning consistency, it does not directly address biases in training data, leaving the risk of perpetuating or amplifying existing societal biases. Additionally, greater reliance on automated reasoning—even when more interpretable—may lead to overconfidence in AI outputs and diminished human oversight. Finally, the substantial computational resources required to train and evaluate large MLLMs raise environmental concerns due to increased energy consumption.

## E  USE OF LARGE LANGUAGE MODELS

We used large language models (LLMs) solely for language editing and polishing purposes. The LLMs were not involved in the design of research ideas, development of methods, data analysis, or interpretation of results. All conceptual contributions, technical content, and scientific claims in this paper are entirely the work of the authors.

