# OpenReview forum: "GRPO-CARE: Consistency-Aware Reinforcement Learning for Multimodal Reasoning"
_ICLR.cc/2026/Conference — ICLR 2026 Conference Withdrawn Submission_

### Official Review · Reviewer_qDNz · 2025-10-29

**Soundness:** 3
**Presentation:** 3
**Contribution:** 3
**Rating:** 6
**Confidence:** 4

**Summary:**

The paper proposes GRPO-CARE, a novel RL algorithm that improves MLLM's multimodal reasoning capabilities by enforcing consistency between the final answer and the generated reasoning trace. In addition, the paper introduces a new validation benchmark, SEED-Bench-R1, that focuses on video understanding that requires intricate visual perception and reasoning/planning capabilities. Using such a benchmark for validation, the paper thoroughly evaluates existing methods and their answer-reasoning trace consistency. Among the experiments, the GRPO-CARE exhibits improvement over other baseline methods on both general benchmark and SEED-Bench-R1.

**Strengths:**

- The paper approaches an important problem: how to improve the model's multimodal reasoning capability by designing better RL algorithms.
- The paper proposes a novel algorithm, GRPO-care, and a new validation benchmark, Seed-Bench-R1, which is beneficial to the general multimodal learning community.
- The motivation is clear, and the considered approach is reasonable, with decent performance improvement over other baseline methods over the proposed Seed-Bench-R1 benchmark.
- The listed and compared baselines are comprehensive, and the ablation study designs were also thoughtful.

**Weaknesses:**

- The performance improvement of GRPO-CARE on the general video understanding benchmark is not significant. Also, the other baselines, such as Video-R1-7B, are not evaluated on Seed-Bench-R1, which makes the reported results slightly less convincing.
- The idea of using a slowly updated EMA reference model for calibrating rollout reasoning trace log probability is interesting, but the paper lacks an in-depth investigation of why this approach works better than other options. A more systematic study on why this works would make the paper more impactful.
- GRPO CARE filtered out low-accuracy trajectories, which degrades its sample learning efficiency and prevents it from effectively learning using negative samples.

**Questions:**

Besides the points in the weakness section, I have the following additional questions:
- How necessary is the trajectory filtering based on the accuracy, and how does this operation affect the final accuracy? For challenging tasks where base models often achieve low accuracy, this could be detrimental.
- What are the accuracy metric and consistency metric used in the experiments (their definition, how to compute them, etc)? Why is the accuracy metric not 0-1?
- How much success of GRPO-CARE can be attributed to the introduced consistency bonus rather than the filtered, high-quality rollouts used in the process?
- How does GRPO-CARE compare with other methods in terms of compute used for training a 7B model, e.g., Video-R1?

---

> ### Author Response · Authors · 2025-11-23
>
> ### **[Q1. Performance improvement on general video benchmarks is not significant. Missing Video-R1-7B on Seed-Bench-R1.]**
>
> We appreciate your insightful feedback. Below, we address your concerns regarding the performance of GRPO-CARE and the baseline comparisons.
>
> **1. Regarding the evaluation of baselines on Seed-Bench-R1:**
>
> We have conducted new experiments by training the base model on Seed-Bench-R1's training data using the Video-R1 algorithm. The results demonstrate that **GRPO-CARE substantially outperforms the Video-R1 baseline**, further validating the effectiveness of our approach.
>
> | Method | L1 | L2 | L3 |
> | :--- | :--- | :--- | :--- |
> | **GRPO-CARE** | **57.0** | **57.0** | **53.4** |
> | Video-R1-7B | 51.4 | 50.9 | 48.1 |
>
> **2. Regarding the performance on general video understanding benchmarks:**
>
> We would like to clarify that GRPO-CARE achieves a **significant improvement over the strong base model, Qwen2.5-VL-7B**, on the general video understanding benchmarks. The primary purpose of comparing our method with Video-R1-7B was to demonstrate that GRPO-CARE can **match or even surpass the performance of current state-of-the-art reinforcement learning-based methods** on these established benchmarks.
>
> **3. On the Simplicity and Efficiency of GRPO-CARE:**
>
> Furthermore, we wish to highlight a crucial advantage of our method: its **simplicity and efficiency**.
>
> *   **High Cost of Video-R1-7B:** The baseline, Video-R1-7B, benefits from an explicit temporal order grounding in its generation process. This approach, while effective, nearly **doubles the RL training cost** and requires a supplementary supervised fine-tuning stage with additional data for model warm-up.
> *   **Efficiency of GRPO-CARE:** In contrast, GRPO-CARE requires a minor modification to the reference model already present in the GRPO framework. This involves just **one additional forward pass** through the reference model per batch for consistency assessment. The EMA update for the reference model adds only minimal training overhead.
>
> This efficiency is confirmed by our training time measurements:
> *   GRPO (baseline): **~12.25 s/iter**
> *   GRPO-CARE (ours): **~12.53 s/iter**
> *   Video-R1-7B: **~23.10 s/iter**
>
> ---
> ### **[Q2: Why does the EMA reference model work better?]**
>
>
> We have provided an ablation in **Table 4** that directly analyzes why the EMA reference model is effective.
>
> **1. EMA provides stronger and improving signals.**
> Training with an EMA-updated reference significantly outperforms using a frozen reference or using the main policy itself, showing that EMA adapts to the evolving reasoning patterns and yields better reward signals.
>
> | Reference Model | L1  | L2  | L3  |
> |-----------------|-----|-----|-----|
> | frozen          | 51.7| 50.4| 51.1|
> | **EMA updated**     | **57.0**| **57.0**| **53.4**|
> | main policy     | 52.7| 53.5| 51.2|
>
>
> **2. EMA does not collapse into the policy.**
> If collapse occurred, its performance would match the “main policy as reference” baseline. Instead, EMA yields consistently higher scores, indicating it remains a distinct and stable reference.
>
> **3. EMA is stable throughout training.**
> A slow update rate allows the reference model to evolve smoothly without drift or reward collapse, unlike a frozen model.
>
> Combined with **Table 3** in our paper, these results show that neither KL-based (e.g., KL-EMA) nor reward-based baselines match the stability and effectiveness of using a **slowly updated EMA reference with group-relative sparse rewards**, validating our design choice.
>
>
> ---
> ### **[Q3: GRPO CARE filtered out low-accuracy trajectories, which degrades its sample learning efficiency and prevents it from effectively learning using negative samples.]**
>
> We clarify that **GRPO-CARE does not filter out low-accuracy trajectories.** As shown in Figure 3 and Algorithm 1, **all trajectories—regardless of accuracy—are used in advantage estimation and policy updates**. The only modification is adding a sparse consistency bonus on top of the standard accuracy reward.
>
> We apply this bonus only to high-accuracy samples to **avoid reinforcing wrong but logically consistent reasoning**, which would otherwise encourage overconfident errors.
>
> For **challenging tasks where base accuracy is low**, **the reward calculation** naturally **reduces to standard GRPO**, because few samples trigger the bonus. Thus, the method does **not** harm sample efficiency or degrade performance in low-accuracy regimes.

---

> ### Author Response · Authors · 2025-11-23
>
> ### **[Q4. Clarification on metrics.]**
>
> **Accuracy Metric:** The metric is adapted to the question format, which is why it is not strictly binary.
> *   **For Multiple-Choice (e.g., SEED-Bench-R1):** We use a **0-1 score**, where 1 is awarded for a correct match with the ground truth.
> *   **For Mixed-Format Data (e.g., Video-R1's training data):** To handle diverse tasks, we follow Video-R1 to use continuous scores. For example, for **Free-form QA**, accuracy is measured by the **ROUGE score**, while for **Regression** tasks, it is calculated as **1 - relative error**.
>
> **Consistency Metric:** This metric evaluates the logical soundness of the model's reasoning process using GPT-4.1 as a judge. The evaluation criteria are:
> 1.  **Logical Soundness:** Does the reasoning logically lead to the answer?
> 2.  **Step-wise Correctness:** Is each step in the reasoning valid and clear?
> 3.  **Error Identification:** Are there any other reasoning fallacies?
>
> The evaluation results are summarized in a structured JSON to produce the final score. **The full prompt and examples are provided in the appendix.**
>
> ---
> ### **[Q5: How much success of GRPO-CARE can be attributed to the introduced consistency bonus rather than the filtered, high-quality rollouts used in the process?]**
>
>
>
>
> First, as we clarified in our response to Q3, **we do not filter out low-quality rollouts**. All generated trajectories contribute to the learning process.
>
> To isolate the impact of our proposed consistency bonus, we compared GRPO-CARE against several strong baselines that also focus on high-accuracy samples in **Table 3**.:
> *   **KL-EMA-HA** and **SepKL-EMA-HA** apply KL regularization only to high-accuracy trajectories.
> *   **DenseCons** and **RefGen** multiply their consistency reward by the accuracy score, effectively weighting the bonus toward correct answers.
>
> | Method                  | L1       | L2       | L3       |
> | ----------------------- | -------- | -------- | -------- |
> | KL-EMA-HA               | 55.1     | 53.8   | 49.2
> | SepKL-EMA-HA | 54.8 |  54.9 | 47.5 |
> | DenseCons  | 56.6  | 55.5  | 50.6|
> | RefGen | 55.2 | 54.2 | 49.4|
> | **GRPO-CARE**               | **57.0**     | **57.0**     | **53.4**     |
>
> As the results show, **GRPO-CARE consistently outperforms all these methods** across every difficulty level. This demonstrates that the performance gain is not merely a byproduct of focusing on high-accuracy samples. Instead, it is the unique design of our adaptive, group-relative consistency bonus that effectively promotes logically sound reasoning, leading to superior overall performance.
>
>
> ---
> ### **[Q6: How does GRPO-CARE compare with other methods in terms of compute used for training a 7B model, e.g., Video-R1?]**
>
>
>
>
> GRPO-CARE is highly computationally efficient. The average training time per iteration for a 7B model is:
>
> *   **Standard GRPO:** ~12.25 s/iter
> *   **GRPO-CARE (ours):** ~12.53 s/iter
> *   **Video-R1:** ~23.10 s/iter
>
> Our method adds a negligible overhead of only **~2.3%** over the GRPO baseline. It is also nearly **twice as fast** as the Video-R1 framework, demonstrating its efficiency and scalability.

---

### Official Review · Reviewer_Rexj · 2025-10-30

**Soundness:** 2
**Presentation:** 2
**Contribution:** 2
**Rating:** 4
**Confidence:** 3

**Summary:**

This paper introduces **SEED-Bench-R1**, a hierarchical benchmark for evaluating post-training methods in multimodal large language models (MLLMs), and proposes **GRPO-CARE**, a consistency-aware reinforcement learning (RL) framework. SEED-Bench-R1 organizes real-world video understanding tasks into three generalization levels (in-distribution, cross-environment, cross-environment-task) to rigorously assess MLLM performance. GRPO-CARE addresses the limitations of outcome-supervised RL (e.g., logical inconsistency between reasoning and answers) by introducing a two-tiered reward system: a base reward for answer correctness and an adaptive consistency bonus derived from an EMA-updated reference model. Experiments demonstrate GRPO-CARE’s superiority over baseline methods, achieving improved accuracy and consistency across SEED-Bench-R1 and transfer tasks.

Key contributions include:

- A structured benchmark (SEED-Bench-R1) for evaluating MLLM post-training methods.
- A novel RL framework (GRPO-CARE) that jointly optimizes answer correctness and reasoning consistency.
- Empirical validation showing GRPO-CARE’s robustness in video understanding and language-only reasoning tasks.

**Strengths:**

**Originality**:

- The hierarchical design of SEED-Bench-R1 addresses a critical gap in evaluating MLLM generalization across controlled OOD scenarios.
- GRPO-CARE’s use of an EMA reference model for likelihood calibration and group-relative sparse rewards is a creative adaptation of existing RL principles to enforce logical consistency.

**Quality**:

- The ablation studies (Tables 3–5) rigorously validate GRPO-CARE’s components (e.g., EMA reference, two-stage filtering).
- Transfer experiments (Tables 6–7) demonstrate the framework’s generalizability to diverse tasks, including language-only reasoning.

**Clarity**:

- The paper is well-structured, with clear explanations of SEED-Bench-R1’s design (Appendix A) and GRPO-CARE’s algorithmic details (Algorithm 1).
- Visualizations (Figures 1–3) effectively illustrate the benchmark’s hierarchy and the framework’s reward mechanism.

**Significance**:

- The work provides actionable insights into the limitations of outcome-supervised RL for MLLMs (e.g., reasoning-answer inconsistency) and offers a practical solution.
- SEED-Bench-R1 fills a need for standardized evaluation in multimodal reasoning, which could benefit future research.

**Weaknesses:**

**1. Limited Benchmark Scope**:

- SEED-Bench-R1 focuses on video understanding but does not cover other critical multimodal domains (e.g., audio-visual tasks, interactive environments). While the authors mention future expansion, the current narrow scope limits the benchmark’s utility for broader MLLM research.

**2. Reliance on GPT-4 for Consistency Evaluation**:

- The consistency metric (Figure 5) depends on GPT-4.1 judgments, which may inherit biases or errors from the LLM. The paper does not validate whether GPT-4.1’s criteria align with human notions of logical coherence, raising concerns about circularity (using LLMs to evaluate LLM-based systems).

**3. Computational Overhead**:

- GRPO-CARE’s two-stage filtering and EMA reference model introduce significant computational costs (e.g., maintaining reference models, group-based calibration). The paper does not quantify these overheads or discuss scalability to larger models (e.g., 70B+ parameters).

**4. Incomplete Comparison to Prior Work**:

- While KL-oriented and reward-based baselines are compared (Table 3), the paper omits recent RL methods like WARP (Ramé et al., 2024) or process-supervised approaches (Luo et al., 2024). A broader comparison would better contextualize GRPO-CARE’s advancements.

**Questions:**

**Key Questions**:

1. **Benchmark Generalizability**: Could SEED-Bench-R1 incorporate additional modalities (e.g., audio) or tasks (e.g., dialogue-based reasoning) to better reflect real-world multimodal challenges?
2. **Consistency Evaluation**: Have you explored alternative consistency metrics (e.g., human evaluation, rule-based checks) to reduce reliance on GPT-4?
3. **Computational Efficiency**: What steps could be taken to reduce GRPO-CARE’s computational overhead (e.g., parameter sharing, dynamic group sizing)?
4. **Broader Baselines**: How does GRPO-CARE compare to WARP or process-supervised RL in terms of consistency and training stability?

**Suggestions**:

- Expand SEED-Bench-R1 to include non-video tasks (e.g., image-text reasoning) to enhance its utility.
- Conduct a sensitivity analysis of GPT-4.1’s consistency judgments against human annotations.
- Provide computational cost metrics (e.g., FLOPs, training time) for GRPO-CARE versus baselines.
- Compare GRPO-CARE to process-supervised methods (e.g., AlphaMath, Lightman et al., 2023) to highlight trade-offs between annotation cost and performance.

---

> ### Author Response · Authors · 2025-11-23
>
> ### **Q1. Limited benchmark scope.**
>
>
> SEED-Bench-R1's primary purpose is to fill a clear gap in existing MLLM post-training evaluations—**no prior benchmark offers large-scale training data with structured hierarchical generalization evaluation that balance visual perception
> and logical reasoning.** Its QA format naturally reflects next-action prediction in an interactive environment, requiring models to reason about evolving visual states and task goals.
>
> We agree that expanding to additional modalities (e.g., audio) and richer interactive settings is a promising direction. Notably, our framework is modality-agnostic and **readily extensible**, as the original video sources used by our benchmark already contain audio tracks and dialogue cues that can be directly incorporated into future iterations of the benchmark.
>
> ---
> ### **Q2. Reliance on GPT-4 for consistency evaluation.**
>
> We acknowledge the concern about relying on GPT-4.1 for consistency evaluation. To ensure reliability, we have already conducted a human validation study: as reported in Appendix B.3 (line 861–863), we randomly sampled 50 GPT-identified consistent and 50 inconsistent cases and compared them with human annotations. The results show a strong Pearson correlation of 0.82, confirming that GPT-4.1’s judgments closely align with human notions of logical coherence.
>
>
> ---
> ### **Q3. Computational overhead.**
>
>
>
> We thank the reviewer for the insightful comments on computational overhead. We respectfully clarify that **the overhead introduced by GRPO-CARE is minimal, not significant**, for the following reasons:
> - **The additional cost is negligible in practice.** The core of our method requires only a single forward pass through the EMA-updated reference model to assess consistency for an entire batch. This cost is trivial compared to the primary computational bottleneck in online RL: autoregressive generation of rollouts from the policy model.
> - **A reference model is a standard component.** Maintaining a reference model is standard practice in algorithms like CRPO, where it is used for KL regularization. Our method leverages this existing component, with the EMA update itself being a near-zero-cost operation. We are not introducing a new, heavy architectural piece.
> - **Empirical results confirm minimal overhead.** We benchmarked the wall-clock time per training iteration. The baseline GRPO takes **12.25s**, while our GRPO-CARE takes **12.53s**. This represents only a minor increase, confirming that the additional computational burden is not a practical concern.
>
> Regarding scalability, although we cannot run full RL on 70B+ models due to resource limits, the overhead of GRPO-CARE scales only with one extra forward pass, not with additional sampling or model duplication. This suggests a similar overhead profile for larger models.

---

> ### Author Response · Authors · 2025-11-23
>
> ### **Q4. Incomplete comparison to prior work.**
>
>
>
> - **Comparison to WARP**
>
> We clarify that our paper **does include** the WARP baseline. In Table 3, the method named **“KL-EMA”** is a WARP-based implementation: we adopt an EMA-updated reference model to compute KL constraints exactly as proposed in WARP. While KL-EMA improves over GRPO, it still falls short of GRPO-CARE:
>
> | Method                  | L1       | L2       | L3       |
> | ----------------------- | -------- | -------- | -------- |
> | GRPO                    | 52.3     | 53.2     | 46.7     |
> | **KL-EMA (WARP-based)** | 54.7     | 54.1     | 49.4     |
> | **GRPO-CARE**           | **57.0** | **57.0** | **53.4** |
>
>
>
> ---
> - **Process-Supervised Approaches**
>
> Process-supervised methods such as AlphaMath rely on step-level annotations or beam-search-derived step rewards. These approaches are **not directly applicable** to general multimodal reasoning, where structured step-by-step supervision is unavailable and is difficult to define.
>
> Nevertheless, we implemented a process-supervised baseline using **open-source LLM judges (Qwen2.5-Instruct-7B)** to explicitly score reasoning consistency. Results show that LLM-judge-based supervision performs significantly worse than GRPO-CARE:
>
> | Method                             | L1       | L2       | L3       |
> | ---------------------------------- | -------- | -------- | -------- |
> | **GRPO-CARE**                      | **57.0** | **57.0** | **53.4** |
> | GRPO + LLM Judge                   | 41.1     | 43.3     | 40.6     |
> | GRPO + LLM Judge (with reflection) | 49.1     | 51.2     | 48.5     |
>
> Even with reflection—which generates additional reasoning tokens to improve judgment—the LLM judge remains clearly inferior.
>
> ---
> - **GRPO-CARE is more effective and efficient**
>
> **Effectiveness.**
> LLM-judge supervision depends on an external evaluator and may introduce noisy or inconsistent reward signals.
> In contrast, our method uses a **self-evolving EMA reference model** initialized from the same weights as the online policy. **This provides stable, gradually improving consistency assessments without requiring a stronger external teacher.**
>
> **Efficiency.**
> GRPO-CARE requires **only one forward pass** through the EMA reference model per batch to compute likelihoods.
> LLM judges must **autoregressively generate tokens**, incurring multiple forward passes; step-level beam search is even more expensive.
> Thus, GRPO-CARE offers substantially lower cost while delivering stronger performance.

---

### Official Review · Reviewer_TZcE · 2025-11-01

**Soundness:** 3
**Presentation:** 3
**Contribution:** 2
**Rating:** 2
**Confidence:** 3

**Summary:**

This paper
1. introduces SEED-Bench-R1, a structured benchmark for evaluating post-training
methods for MLLM
2. proposes a RL framework: GRPO-CARE for MLLMs to jointly optimizes for answer's correctness and logical consistency.
3.  demonstrates  that GRPO-CARE consistently outperforms standard GRPO for experiments on SEED-Bench-R1

**Strengths:**

The paper identifies an important direction of improving multimodal reasoning capabilities through consistency reward. However the paper could be strengthened by more experiments and analysis as mentioned below.

**Weaknesses:**

1. The paper could be strengthened by providing analysis on how the weighting between correctness reward and consistency reward could affect model performance
2. The vast majority of the paper's experiments, analyses, and conclusions are based on a single model architecture : Qwen2.5-VL. The effectiveness of the proposed method could be further tested on other model architectures and scales.

**Questions:**

NA

---

> ### Author Response · Authors · 2025-11-23
>
> ### **Q1. Analysis on the weighting between correctness reward and consistency reward.**
>
>
> We conducted an ablation study on the consistency reward coefficient ($λ_cons$) to analyze its impact on performance.
>
> | $λ_{cons}$ | L1 | L2 | L3 |
> | :--- | :---: | :---: | :---: |
> | 0.1 | 54.2 | 55.5 | 51.1 |
> | 0.3 | 56.7 | 55.9 | 50.3 |
> | 0.4 | 56.4 | 55.3 | 53.1 |
> | **0.5** | **57.0** | **57.0** | **53.4** |
> | 0.6 | 55.8 | 54.7 | 51.6 |
> | 0.7 | 54.0 | 52.3 | 50.6 |
> | 1.0 | 55.6 | 55.9 | 53.1 |
>
> The results show that our method is robust to this hyperparameter, achieving strong performance around the value of 0.5.
>
> We used $λ_{cons} = 0.5$ for all experiments, which confirms it is a stable and effective choice that does not require per-benchmark tuning.
>
> ---
> ### **Q2. The vast majority of the paper's experiments, analyses, and conclusions are based on a single model architecture.**
>
>
> Our choice of the Qwen model series is **aligned with established practices in the MLLM post-training community.** Leading works (e.g., Visual-RFT, Video-R1, OpenVLThinker) and popular open-source RL frameworks (e.g., EasyR1, R1-V) have widely adopted Qwen models due to their SOTA performance and strong open-source support, making them a robust baseline for evaluating new methods.
>
> Furthermore, to validate the robustness of our approach, we did conduct experiments on other model variants. As detailed in **Appendix B.2** of our submission, we evaluated GRPO-CARE on **Qwen2-VL-7B** (a different version) and **Qwen2.5-VL-3B** (a different scale). For convenience, we have included these results below:
>
> | Model | L1 | L2 | L3 |
> | :--- | :---: | :---: | :---: |
> | Qwen2-VL-7B | 34.7 | 34.0 | 31.6 |
> | Qwen2-VL-7B + SFT | 43.8 | 44.1 | 38.2 |
> | Qwen2-VL-7B + GRPO | 46.0 | 50.2 | 44.9 |
> | **Qwen2-VL-7B + GRPO-CARE** | **57.2** | **56.2** | **53.8** |
> | Qwen2.5-VL-3B | 31.3 | 32.7 | 28.2 |
> | Qwen2.5-VL-3B + SFT | 35.9 | 39.1 | 33.7 |
> | Qwen2.5-VL-3B + GRPO | 39.6 | 41.0 | 35.4 |
> | **Qwen2.5-VL-3B + GRPO-CARE** | **47.1** | **48.8** | **43.5** |
>
> The results clearly show that **GRPO-CARE consistently and significantly outperforms both SFT and standard GRPO across different model versions and scales**.
>
>
>
>
> ---
> ### **Q3. Clarification on the broader contributions of this work.**
>
>
> We sincerely thank the reviewer for recognizing the importance of our work on improving multimodal reasoning with a consistency reward. But we would like to respectfully highlight that our work is built on three synergistic contributions, **as recognized by Reviewer `Rexj`, `qDNz`**:
>
>
> 1.  **A Structured Benchmark:** We introduced **SEED-Bench-R1**, a rigorous benchmark for evaluating MLLM post-training methods with a novel hierarchical design.
> 2.  **A Core Problem Analysis:** We provided a systematic analysis illuminating a key limitation of outcome-supervised RL in preserving logical coherence
> 3.  **A Novel and Efficient Method:** We proposed **GRPO-CARE**, a novel and elegant RL framework that delivers robust gains in both accuracy and consistency. It utilizes an EMA-based reference model to provide self-supervised signals for consistency assessments without requiring a stronger external teacher.
>
> **Our current submission provides a comprehensive set of experiments to validate these claims.** For instance, **Reviewer `Rexj`** recognizes that the ablation studies (**Tables 3–5**) rigorously validate GRPO-CARE’s components, while the transfer experiments (**Tables 6–7**) demonstrate its generalizability to diverse video and even language-only reasoning tasks. **Reviewer `qDNz`** acknowledges that the listed and compared baselines are comprehensive, and the ablation study designs were also thoughtful.
>
> We fully agree with the reviewer that the paper can always be strengthened with more analysis. We commit to incorporating additional experiments in the final version to further solidify our findings.
>
> **We hope the reviewer will consider the full scope of our contributions—the benchmark, the problem analysis, and our extensively validated method, in shaping the final rating and justification.**

---

### Note · Authors · 2026-01-05

I have read and agree with the venue's withdrawal policy on behalf of myself and my co-authors.